# Maternal Socioeconomic Status and the Initiation and Duration of Breastfeeding in Western Europe Versus Southern Africa: A Systematic Review—A Contribution from the ConcePTION Project

**DOI:** 10.3390/nu17060946

**Published:** 2025-03-08

**Authors:** Martje Van Neste, Katoo Verschoren, Rani Kempenaers, An Eerdekens, Danine Kitshoff, Karel Allegaert, Annick Bogaerts

**Affiliations:** 1Clinical Pharmacology and Pharmacotherapy, Department of Pharmaceutical and Pharmacological Sciences, KU Leuven, 3000 Leuven, Belgium; martje.vanneste@kuleuven.be; 2Leuven Child and Youth Institute, KU Leuven, 3000 Leuven, Belgium; annick.bogaerts@kuleuven.be; 3Department of Public Health & Primary Care, Academic Centre for Nursing and Midwifery, 3000 Leuven, Belgium; katoo.verschoren@student.kuleuven.be (K.V.); rani.kempenaers@hotmail.com (R.K.); 4Neonatal Intensive Care Unit, University Hospitals Leuven, 3000 Leuven, Belgium; an.eerdekens@uzleuven.be; 5Department of Nursing and Midwifery, Faculty of Medicine and Health Sciences, Stellenbosch University, Cape Town 7602, South Africa; danenek@sun.ac.za; 6Department of Development and Regeneration, KU Leuven, 3000 Leuven, Belgium; 7Department of Hospital Pharmacy, Erasmus University Medical Center, 3000 CA Rotterdam, The Netherlands; 8Faculty of Health, University of Plymouth, Devon PL4 8AA, UK

**Keywords:** socioeconomic status (SES), socioeconomic factor, social inequality, global health, Western Europe, Southern Africa, breastfeeding, lactation, initiation, duration

## Abstract

Breastfeeding is associated with many health benefits, while its prevalence is determined by numerous factors, including socioeconomic status (SES). SES is the position of an individual on the socioeconomic scale, using occupation, education, income, place of residence, and wealth as key indicators. Since its interrelationship with health is complex, world region-specific insights into the relevant socioeconomic inequalities impacting breastfeeding practices are crucial to effectively address these. The purpose of this systematic review is, therefore, to explore SES indicators affecting breastfeeding initiation and duration in two different United Nations-defined regions, Western Europe and Southern Africa to assess (dis)similarities, as these can guide region-specific, targeted interventions to improve practices. A systematic literature search was conducted across seven databases, of which 47 articles were included. The risk of bias was assessed, and outcome data related to SES as well as breastfeeding initiation and duration were collected. Higher education consistently leads to better breastfeeding initiation outcomes, but economic constraints and employment in informal sectors hinder breastfeeding practices in Southern Africa. In Western Europe, supportive working conditions and a migration background have a positive impact, while employment status and income show rather mixed effects. Community, regional, and religious factors play significant, ambiguous roles. In South Africa, food insecurity, the living environment, and geographic location complicate breastfeeding. This systematic review highlights the significant influence of SES on breastfeeding initiation and duration in Western Europe and Southern Africa, while the specific factors indeed vary between both regions. This systematic review therefore illustrates the relevance of region-specific SES factors, impacting breastfeeding practices. Addressing these barriers with region-specific, targeted approaches may result in substantial progress toward achieving global breastfeeding goals. Registration: PROSPERO (CRD42023473433).

## 1. Introduction

Socioeconomic status (SES) is the position of an individual on the socioeconomic scale, with occupation, education, income, place of residence, and wealth as key indicators [1]. Its effect on health is complex and requires more research, since SES both influences and is influenced by variables including lifestyle, social inclusion, medical care, nutrition, and psychological factors [2,3]. All underlying determinants are strongly correlated, with some factors occurring more frequently at lower status levels and others at higher status levels [2,4,5]. It is therefore reasonable to explore how and to what extent these SES determinants have a world region-specific impact and relevance.

Initially, SES was described as low, medium, and high, but there appears to be a rather more gradual relationship. The social gradient is associated with health outcomes, e.g., people who are less advantaged have worse health than those who are more advantaged in terms of social position [6,7]. According to a study conducted in the United States by Foster et al. (2023), women with lower SES initiate breastfeeding less frequently than women with higher SES [8]. Women with higher SES, for example, find themselves more frequently in the position to stay at home to take care of their children, which results in more time for exclusive breastfeeding [9].

Besides SES, ethnicity is recognized as an important determinant influencing the initiation and duration of breastfeeding. For example, research highlights a difference of 17% between different ethnicities in the United States, although disparities were often state-specific [10]. Moreover, ethnicity is correlated with socio-cultural environments, which highlights the complex interplay of different indicators [11]. Comparative studies across different regions help to identify similarities and differences in SES variables [12]. By addressing socioeconomic inequalities impacting health in relation to the regional setting, health outcomes for the global population can be improved [3,4,13,14].

The World Health Organization (WHO) recommendations advocate exclusive breastfeeding for the first six months of life and partial breastfeeding up to two years of age [15]. Human milk improves child health and survival and ensures health benefits. For instance, it is safe to drink and contains nutrients and antibodies, that are important against infections. Children who are breastfed score better on intelligence tests and are less likely to become obese or suffer from diabetes later in life. Long-term benefits for the mother, as Binns et al. (2016) found in their systematic review, include—amongst others—a reduced risk of obesity, type 2 diabetes, and heart disease [16]. These benefits often occur or become more prominent with an extended duration of breastfeeding following exclusive breastfeeding for the first six months.

Breastfeeding initiation is the actual start of breastfeeding and is recommended within one hour after birth, with prevalence ranging from 11.4% in Saudi Arabia to 83.3% in Sri Lanka [17]. Despite the breastfeeding recommendations, the WHO published in 2023 that only 48% of infants less than six months old worldwide are exclusively breastfed [15,18]. Fortunately, progress is happening across different regions, with 22 countries in Africa, Asia Europe, and Oceania reporting significant increases (>10%). South Asia and Eastern and Southern Africa overperform (>48%), while East Asia, the Pacific, the Middle East, and North America underperform (<48%) on exclusive breastfeeding rates at 6 months [18,19,20].

However, only 32% of all South African newborns were fed exclusively with human milk for the first six months according to the South African Demographic Health Survey in 2016, making this the country with the lowest breastfeeding rates in Africa [19]. The WHO has set six global targets for 2025 to improve maternal, infant, and young child nutrition, including raising the exclusive breastfeeding rate during the first six months to at least 50% [20]. To attain these goals, systematic information on the regional impact of SES on breastfeeding practices is needed to identify intervention targets that will be effective for the specific population. Research has been conducted on how SES affects the initiation and duration of breastfeeding in different regions, including Western Europe and Southern Africa. However, there has not yet been a comparative analysis of both regions, even though such information could be instrumental in developing region-specific targeted interventions.

This review is situated within the ConcePTION project. This project, initiated by the Innovative Medicine Initiative (IMI), is a European public–private partnership that aims to generate evidence-based information on lactation and medicine exposure [21]. As the benefits of breastfeeding are numerous and can outweigh the risks of medication exposure via human milk, interventions to improve breastfeeding rates should be promoted [22].

The purpose of this systematic review is, therefore, to determine the relationship between maternal SES and breastfeeding initiation and duration in Western Europe versus Southern Africa to identify intervention targets and possible obstacles to improving breastfeeding practices. Southern Africa and Western Europe represent contrasting settings in terms of SES distribution, healthcare infrastructure, and breastfeeding promotion initiatives. By examining these two regions, our aim is to highlight both universal trends and context-specific challenges in breastfeeding practices influenced by SES. Understanding these dynamics can contribute to a more nuanced region-specific and global perspective on the interplay between SES and breastfeeding. In line with the WHO definition of SES, we explore the following factors in both regions: educational level; employment, income, and the work environment; housing, basic amenities and the environment; food insecurity; origin and migration; ethnicity and cultural aspects [5].

## 2. Materials and Methods

This systematic review was registered on PROSPERO (CRD42023473433, 28 October 2023) and was conducted following the guidelines of the “Preferred Reporting Items for Systematic Reviews and Meta-Analyses” (PRISMA) (see Appendix A) [23].

This systematic literature search was performed in the following databases: PubMed, Embase, CINAHL, Web of Science, Cochrane library, ProQuest, and Scopus. Key concepts included (i) Southern Africa and Western Europe according to the official United Nations geoscheme, (ii) initiation and duration of breastfeeding, and (iii) socioeconomic factors (see full search strategy in Appendix A provided). Germany, Liechtenstein, Luxembourg, Austria, Switzerland, Belgium, France, Monaco, and the Netherlands were included as countries for Western Europe, and Botswana, Eswatini, Lesotho, Namibia, and South Africa for Southern Africa [24].

Studies had to be reported in English or Dutch on breastfeeding initiation and/or duration after normal pregnancy and birth. Furthermore, the articles had to focus on one of the SES factors according to the definition of the WHO: income and/or social protection; education; unemployment and job insecurity; working life conditions; food insecurity; housing, basic amenities, and the environment; social inclusion and non-discrimination; structural conflict and access to affordable health services of decent quality [5].

Exclusion criteria comprised preterm birth (<37 weeks of gestational age), mothers aged >40 years old, complications during pregnancy or birth, maternal chronic illnesses and relevant medical history, multiple gestation, and substance abuse (drugs/alcohol/nicotine). Studies written before 2008 were excluded to ensure the focus remains on recent publications, reflecting the most current influences of SES on health and WHO guidelines regarding infant and young child feeding [25]. Lastly, case series, case reports, and secondary articles were excluded from this review.

The literature search was carried out on 21 December 2023, after which duplicates were removed according to Bramer et al. (2016) [26]. Articles were imported into Rayyan™ software(version 1.5.5.) for screening. First, articles were screened blindly on title and abstract by two independent researchers (K.V. and R.K.). Afterward, the remaining articles were screened blindly based on full text by the same two independent researchers (K.V. and R.K.). In case of conflict, a third independent reviewer (M.V.N.) was involved to reach the final decision (66.6% consensus). The searches were reconducted prior to the final analysis on 27 June 2024.

The risk of bias in randomized controlled trials was assessed by the Risk of Bias 2 (RoB2) tool and the quality assessment of case-control; cohort and cross-sectional studies were considered using the Newcastle–Ottawa Scale (NOS) [27,28]. The Qualitative Assessment and Review Instrument (QARI) was used for the critical appraisal of qualitative studies [29]. No studies/articles were excluded based on their quality of evidence, yet the scored criteria were considered during synthesis and analysis. This step was conducted independently by two researchers (K.V. and R.K.) with a third reviewer (M.V.N) in case of conflict. Data extraction, using a data extraction form in Microsoft Excel (version 2406), was carried out by the same researchers. This search strategy was reconducted on 27 June 2024, prior to the final analysis, to account for new articles. A narrative synthesis was provided on SES factors related to breastfeeding initiation and duration outcomes for both Western Europe and Southern Africa.

## 3. Results

### 3.1. Characteristics of the Included Studies

In the search, 5106 articles were retrieved from online databases, of which 3129 were duplicates and were removed. Screening by title and abstract of the 1977 remaining articles resulted in 133 articles for full-text screening. Of these, 86 articles were excluded, mainly for outcomes (n = 39) or language (n = 19). Finally, 47 articles were included (Figure 1).

When this search strategy was reconducted (June 2024), this resulted in 127 new studies, but no new articles were included.

The quality assessment of the 47 included articles was primarily conducted using the NOS (n = 41; 87%). The risk of bias was assessed to a lesser extent using the QARI tool (n = 5; 11%) and the RoB2 tool (n = 1; 2%). Most articles were of good quality (n = 35; 74%). Some articles were of very good quality (n = 6; 13%) and a similar number were of satisfactory quality (n = 6; 13%). No articles with insufficient quality were included (see Appendix A).

Studies reported on eight different countries. For Western Europe, studies focused on France (n = 12; 26%) [30,31,32,33,34,35,36,37,38,39,40,41], Germany (n = 7; 15%) [42,43,44,45,46,47,48], the Netherlands (n = 5; 11%) [49,50,51,52,53], Austria (n = 2, 4%) [54,55], and Belgium (n = 2, 4%) [56,57]. Additionally, there was one study that conducted a comparative analysis between France and Germany (n = 1; 2%) [58]. For Southern Africa, studies were retrieved from those reporting South Africa (n = 16, 34%) [59,60,61,62,63,64,65,66,67,68,69,70,71,72,73,74], Namibia (n = 1, 2%) [75], and Eswatini (n = 1, 2%) [76].

Some cohorts were studied in multiple articles, more specifically the EDEN mother–child cohort [31,32,38], the French national birth cohort ELFE [33,34,40,41], and the Generation R Study group [49,50,51]. Furthermore, Kohlhuber et al. (2008) and Rebhan et al. (2009) reported on the same cohort [43,48], as did Robert et al. (2014a) and Robert et al. (2014b) [56,57].

### 3.2. Socioeconomic Factors

Tables summarizing the findings for each SES factor according to the WHO for breastfeeding initiation and duration separately can be found in the Appendix A [5].

#### 3.2.1. Educational Level

Studies in Western Europe consistently demonstrated that maternal education level is a crucial factor for better breastfeeding practices, both initiation and duration [31,32,33,34,36,37,38,39,40,43,44,46,47,48,50,51,54,55,56]. A comparative study in France found that 55% of the mothers without a diploma started breastfeeding, compared to 83.4% of the mothers with a university degree (*p* ≤ 0.001) [38]. Similarly, a Dutch cohort study revealed 73.1% of the mothers with lower education initiated breastfeeding, compared to 95.5% of the mothers with higher education (*p* < 0.001) [50]. A cohort study in Germany reported increased breastfeeding rates over an eleven-year period among mothers who finished at least 12 years of education, while a similar change was lacking for less educated mothers [46].

The limited studies discussing Southern Africa’s breastfeeding initiation rates demonstrated a positive relationship with maternal educational level [60,69,72,75]. However, the effect on breastfeeding duration was ambiguous [68,76], and one study reported no relationship between breastfeeding duration and educational level in South Africa [68].

#### 3.2.2. Employment, Income and the Work Environment

Sixteen studies were considered for Western Europe [32,33,34,35,36,38,39,40,44,46,47,53,54,56,57,58]. Maternal employment at birth and return-to-work time did not significantly affect the initiation of breastfeeding [32,35,38]. However, its duration was significantly influenced by return to work within the first year postpartum, with lower rates of breastfeeding continuation among women re-entering the workforce [34,35,40,47,54,56,57,58]. When comparing two European cohorts, return to work was one of the main reasons to cease breastfeeding in the first 3 months postpartum for both German and French women (27% and 19% of reasons, respectively) [58]. Nevertheless, these findings were contradicted in other Western European studies [36,38,46]. Workplace interventions and policies to create a breastfeeding-supportive environment, such as flexible hours, had a positive impact on breastfeeding duration [35,53,54]. Furthermore, in a single country-specific study, higher income was associated with the intention of partial breastfeeding, while an average income was correlated with exclusive breastfeeding motivation [44]. However, no consistent association was observed between breastfeeding duration and income [38,54,57].

Studies on Southern African countries reported conflicting findings [59,60,61,63,65,66,67,68,70,71,72,73,74,75,76]. Employment was not linked to breastfeeding initiation, but return to work had a substantial impact on breastfeeding duration, as unemployed mothers were 55% less likely to cease breastfeeding before six months [76]. Breastfeeding initiation was higher in women who gave birth in hospitals with baby-friendly programs, which was correlated with higher income [66]. Contrarily, a study in Namibia in 2000 showed 82% higher odds of breastfeeding initiation in poorer households [75]. Breastfeeding cessation was particularly pronounced among informal workers with low income, such as domestic workers or informal traders, who face unique obstacles at their employment [73]. Additionally, they are forced to return to work due to financial imperatives [61]. Furthermore, they often face environments that lack support for breastfeeding, have insufficient childcare, and contribute to work-related stress [59,63,70,71,73]. However, some studies reported no or weaker correlations between breastfeeding and employment status [60,65,68].

#### 3.2.3. Housing, Basic Amenities, and the Environment

Despite the daunting challenges of homelessness, 86% of mothers in homeless populations in France started breastfeeding, and 59% continued up to 6 months or more. This trend highlights the resilience of breastfeeding initiation in this population with restricted access to resources and unstable living conditions [37]. However, regional disparities were reported in Germany and France, emphasizing that contextual factors have a significant influence on breastfeeding behavior [36,45]. For instance, Libuda et al. (2014) found higher exclusive breastfeeding rates at 4 months in large communities (>100,000 citizens) in Germany, but this result was not confirmed in other studies [45].

Similarly to Western Europe, geographical and regional disparities were reported. In rural areas of South Africa, early initiation of breastfeeding is 42% less likely than in urban settings, e.g., in Eastern Cape and KwaZulu-Natal, due to varying access to basic services and limited healthcare facilities and support services [62,75]. In urban settings, on the other hand, breastfeeding is perceived as impractical due to stress from living in overcrowded and substandard housing among migrants in Cape Town [70]. This stress was augmented by a lack of safe environments, limited access to transportation, hygiene concerns, infrastructural challenges, and high levels of crime and gender-based violence [61,71]. Therefore, urban mothers were also less likely to continue breastfeeding after 14 weeks postpartum [68,72]. As formula feeding was often unsafe or unsustainable given the unhygienic conditions and lack of clean water, breastfeeding was still the first infant feeding method (72% of mothers) in KwaZulu-Natal [60]. This also resulted in a small decrease in the probability of discontinuing exclusive breastfeeding in socially vulnerable mothers [76]. On the other hand, mothers working as domestic workers often live with their employers, which results in access to safe water and sanitation [73]. However, these mothers encountered unique challenges regarding childcare [73].

#### 3.2.4. Origin and Migration

Overall, mothers with a migration background had higher breastfeeding rates and longer breastfeeding durations than mothers with no migration background in Western Europe [32,33,34,35,37,38,39,40,41,42,44,57]. A longitudinal study in France reported that 66.7% of mothers in the majority population breastfed their child at birth compared to 75.9% of mothers who are descendants of immigrants and 88.2% of mothers who are immigrants [40]. Furthermore, this study also described that the country of origin influences breastfeeding practices, which was confirmed in a French study in homeless families [37,40]. For instance, 55% of mothers from Sub-Saharan Africa living in France breastfed for up to six months as opposed to 40% of immigrants from other countries [40]. However, this effect was only observed for any breastfeeding practice as the maternal country of birth did not have an influence on predominant or exclusive breastfeeding [41]. Additionally, the reason for migration was a relevant factor as well, since migration due to violence was associated with shorter breastfeeding durations [37].

Little was known about the impact of origin and migration status in Southern Africa as only one study reported that migration status affected breastfeeding practices due to the impact of an unknown environment [70].

#### 3.2.5. Ethnicity and Cultural Aspects

Six studies reported on the impact of ethnicity and cultural aspects on breastfeeding practices in Western Europe [30,37,38,41,49,52]. Cultural values and practices from non-European origin countries positively influenced the initiation rates of breastfeeding and its duration among foreign-born mothers [41]. However, a Dutch study concluded that despite more mothers from ethnic minorities (Caribbean or Mediterranean) initiating breastfeeding, they were also more likely to discontinue [49]. Additionally, religion played a significant role in breastfeeding initiation rates in Western Europe, e.g., the proportion of Catholics is negatively correlated with breastfeeding initiation and the proportion of Protestants is positively correlated [30].

One study in Southern Africa described that differences in societal norms, cultural norms, and traditional beliefs affect the effort to breastfeed, both in a positive and a negative way [63].

#### 3.2.6. Access to Affordable Health Services of Decent Quality

Except for one article stating that access to health services for informal workers in KwaZulu-Natal was comparable to the general South African population, no information was found for Southern Africa or Western Europe [73]. Horwood et al. (2019) described a positive impact of advice and support from health workers on breastfeeding practices, provided the health workers are aware of the context and challenges of informal workers [73].

#### 3.2.7. Food Insecurity

The link between poor diets and low breastfeeding rates highlighted how nutritional deficiencies might undermine maternal and child health practices, according to Horwood et al. (2019) [73]. This underscored the need for more affordable and accessible nutritional options for mothers and infants in these Southern African regions to support breastfeeding and overall health. Conversely, the perceived high cost of formula feeding frequently prompted mothers to breastfeed, especially in financially constrained settings that cause limited access to alternative nutrition sources [64]. On the other hand, the lower cost of formula feeding in different contexts, e.g., Cape Town, combined with the “culture of formula” might have contributed to lower breastfeeding rates [70]. No data were available for Western Europe.

## 4. Discussion

This systematic review investigated the relationship between SES and the initiation and duration of breastfeeding. This is the first review to compare these breastfeeding practices between Western Europe and Southern Africa. Common and unique variables were identified to understand and possibly improve global breastfeeding rates and, ultimately, child and maternal health outcomes. An overview of the results is shown in Table 1.

This review found that SES significantly impacts breastfeeding practices in both Western Europe and Southern Africa, albeit in different ways. In Western Europe, higher maternal education, supportive working conditions, and a migration background were consistently associated with higher breastfeeding initiation and longer duration. Employment status shows mixed effects on breastfeeding duration as income remains an ambiguous factor. Community and regional factors, along with religious beliefs, also play significant roles in breastfeeding practices.

These findings are in line with the previously discussed significant impact of SES factors on breastfeeding behaviors in Canada, particularly noting higher rates of early breastfeeding cessation among socioeconomically marginalized populations [77]. Similarly, Alvarez-Galvez et al. (2016) demonstrated that higher SES is correlated with better health outcomes, including increased breastfeeding rates [2]. These findings were further supported in Western Europe as higher education is associated with improved breastfeeding practices [78,79]. As for income, our results are ambiguous, which reflects the existing literature. A correlation between higher maternal earnings and higher rates of breastfeeding duration has been reported [80]. On the other hand, the opposite has been discussed as well, namely that continued breastfeeding is more common in poor mothers than in wealthy mothers, even though rates seem to be dropping among poor mothers while remaining stable in rich mothers [81].

Informal employment, food insecurity, and living environment were prominent barriers to breastfeeding practices in Southern Africa. Employment in informal sectors and economic constraints significantly hinder breastfeeding practices as many mothers return to work soon after birth due to financial necessity. Food insecurity further complicates breastfeeding. On the other hand, there was no association observed between water access or water-fetching and (exclusive) breastfeeding rates [60,82]. Furthermore, geographic location plays a crucial role in Southern Africa, with rural areas facing greater challenges. A migration background is associated with higher breastfeeding initiation rates in Southern Africa, but these associations should be further studied using more in-depth qualitative designs.

The discrepancy between Western Europe and Southern Africa was mainly observed regarding education level. Whereas maternal education strongly influences the initiation and duration of breastfeeding in Western Europe, the effect of education is more nuanced in Southern Africa. Higher education levels are positively associated with breastfeeding initiation. However, this relationship is unclear when it comes to breastfeeding duration. As knowledge is limited, data on religion and cultural aspects, access to health services, and food insecurity could not be compared between both regions.

Multiple successful interventions promoting breastfeeding have been implemented around the world, such as the Baby-Friendly Hospital Initiative (BFHI) and Baby-Friendly Community Initiative (BHCI) [83]. These initiatives successfully promote and support breastfeeding practices in maternity wards and communities [83,84]. Several potential key interventions were identified in this review when promoting and supporting breastfeeding in Western Europe and Southern Africa in the future. In Western Europe, targeted programs should focus on providing essential resources and support for lower-educated mothers, such as workshops, informational materials on the benefits of breastfeeding, and personalized assistance from healthcare professionals. Moreover, it is important to ensure that all mothers, regarding income level, have access to breastfeeding resources and support, for instance by expanding the BFHI and BFCI [83,84]. Breastfeeding promotion materials that respect and incorporate religious values could improve breastfeeding rates for religious populations, similar to the United States [30,85]. In Southern Africa, interventions that provide financial support to mothers, particularly those working in informal sectors, e.g., paid maternity leave, could be implemented to reduce the necessity of returning to work soon after childbirth [63,83]. Additionally, programs targeting mothers with a lower education to give essential support could improve breastfeeding practices, similar to Western Europe. Moreover, breastfeeding support for urban mothers should address the unique challenges of overcrowding and substandard housing conditions. By coordinating efforts of governments, non-governmental organizations, and communities, significant improvements in breastfeeding practices could be achieved, leading to better health outcomes for both mothers and children [86].

While conducting this systematic review, some limitations were identified. Articles on the Southern African countries were limited, especially in certain countries, and were overrepresented for South Africa, resulting in potential within-region bias. Furthermore, only English and Dutch articles were included in this review, which may have resulted in missing relevant articles from non-English- or non-Dutch-speaking countries. Additionally, since there was a high variability in population and study designs, direct comparison of Western Europe and Southern Africa was challenging. For example, the prevalence of Human Immunodeficiency Virus (HIV) is higher in African regions and impacts the decision to initiate breastfeeding in HIV-infected women [87,88]. Last, some studies reported on overlapping cohorts (e.g., Etude Longitudinale Française depuis l’Enfance (ELFE) study), which might have resulted in inflating the influence of certain cohorts [33,34,40,41].

The strengths of this review include the systematic search covering multiple databases, ensuring a broad inclusion of studies. Moreover, studies were rigorously assessed for quality using the RoB2 tool, NOS, and QARI, providing a robust evaluation of study reliability. Finally, this review included studies with diverse study designs, covering cohort, cross-sectional, and qualitative studies, which further enriched the data synthesis.

The insights of this review underscore the critical role of socioeconomic factors in breastfeeding practices, emphasizing the relevance and the need for targeted interventions to globally support breastfeeding across diverse socioeconomic contexts. However, further qualitative research on the ambiguous effects of certain socioeconomic factors should deepen the understanding of the relationships with breastfeeding, more specifically on income and employment in Western Europe and migration status, ethnicity, and cultural practices in Southern Africa. This research is needed to further develop evidence-based interventions to address the current challenges and to tailor interventions to the specific needs of different communities.

## 5. Conclusions

This systematic review highlights the substantial association of SES and breastfeeding initiation and duration in both Western Europe and Southern Africa, though the specific factors and their relation vary between these regions. In Western Europe, breastfeeding rates are increased with higher maternal education, supportive working conditions, and a migration background. Breastfeeding rates in Southern Africa were negatively affected by informal employment, food insecurity, and living environment. This systematic review therefore illustrates the relevance of region-specific SES factors, impacting breastfeeding practices.

Understanding the complex interplay between SES and breastfeeding practices can inform targeted interventions to improve child and maternal health outcomes across or tailored to different regions. Addressing these barriers with region-specific, targeted approaches, significant and impactful strides can be made toward achieving global breastfeeding goals.

## Figures and Tables

**Figure 1 nutrients-17-00946-f001:**
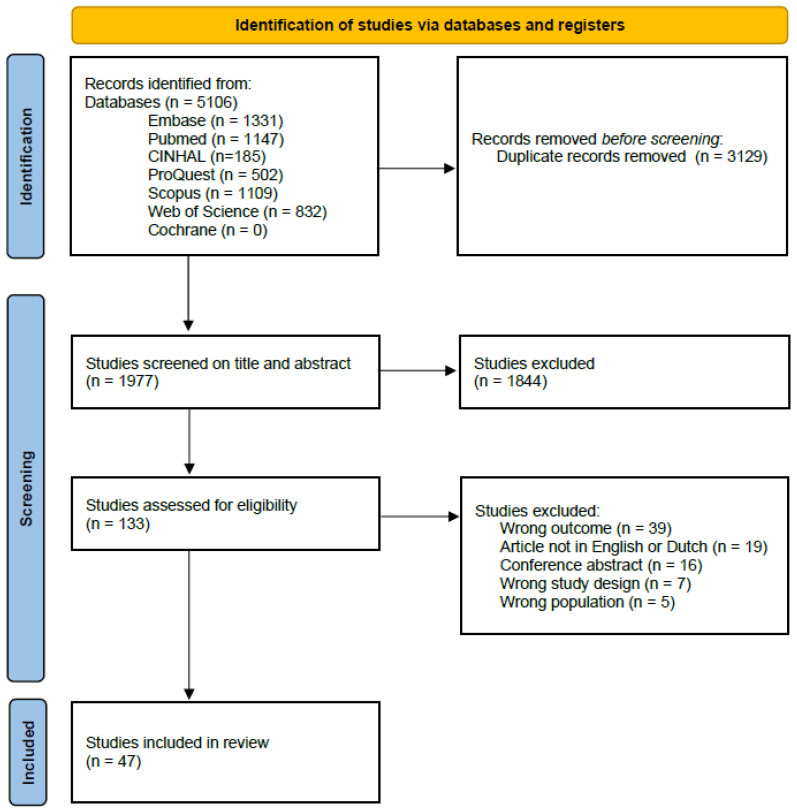
PRISMA 2020 flow diagram for study selection and inclusion [23].

**Table 1 nutrients-17-00946-t001:** Overview of results.

SES Determinant	Western Europe	Southern Africa
Educational level	Higher maternal education is consistently leading to better breastfeeding initiation and duration outcomes.	For breastfeeding initiation, limited studies about the determinant “educational level” revealed a positive relationship. There was no relationship between breastfeeding duration and educational level, but only two studies reported on this subject [68,76].
Employment, income, and the work environment	The results for the determinants “income” and “employment” are ambiguous or conflicting. Some studies described a positive relationship between income and breastfeeding initiation and duration, while other studies found a negative relationship [38,39,54,57]. Maternal employment at delivery and return to work did not significantly influence breastfeeding initiation [32,38]. Flexible working hours and specific job types seem to support continued breastfeeding in Western Europe, emphasizing the role of workplace interventions and policies regarding nursing leave in creating a breastfeeding-supportive environment [35,53,54].	Higher income in Southern Africa is associated with higher breastfeeding initiation rates and lower breastfeeding duration rates, but the results are conflicting [66,72]. Unemployed mothers were less likely to stop exclusive breastfeeding before six months in comparison to employed mothers [76]. Informal working mothers often find themselves in workplace environments unsupportive of breastfeeding, resulting in lower breastfeeding duration rates within this employment category [63,71,73]. Return to work significantly influences breastfeeding duration rates negatively in Southern Africa.
Housing, basic amenities, and the environment	There is an impact of homelessness on breastfeeding initiation and duration rates, as a notable proportion of homeless mothers in Western Europe are initiating breastfeeding. Furthermore, the majority of mothers continue breastfeeding for up to six months or more [37]. A regional impact on breastfeeding initiation was observed [36,45]. Furthermore, mothers in smaller communities were more likely to prolong breastfeeding duration [45].	Geographic location significantly influences breastfeeding initiation and duration, e.g., in rural areas of South Africa, reduced odds for early initiation of breastfeeding were found, highlighting persistent challenges in achieving timely breastfeeding initiation across regions [75]. In urban settings, the stress of living in overcrowded and substandard housing contributes to perceptions that breastfeeding is impractical leading to lower breastfeeding initiation and duration rates [70]. Contradictorily, given the unhygienic conditions and lack of clean water, formula feeding is often neither safe nor sustainable. Therefore, breastfeeding is still chosen as the first infant feeding method [60].
Origin and migration	There is a clear, positive relation between mothers with a migration background and breastfeeding initiation and duration rates in Western Europe [37,38,40,41,42].	There is a clear positive relationship between mothers with a migration background and breastfeeding initiation [70]. No studies investigated the impact of migration on breastfeeding duration in Southern Africa.
Ethnicity and cultural aspects	Religion plays a significant role in the differences in breastfeeding initiation rates in Western Europe, with the direction of the relation depending on the type of religion [30].	No results can be discussed regarding the relationship between religion and breastfeeding initiation and duration.
Access to affordable health services of decent quality	No information was found regarding access to health services in Western Europe.	Only one article specifically addresses access to affordable health services of decent quality without discussing its effect on breastfeeding practices. Horwood et al. (2019) merely stated that access to health services for informal women workers in KwaZulu-Natal is good and comparable to the general population in South Africa [73]. No information was found regarding access to health services for the rest of Southern Africa.
Food insecurity	No data on the determinant “food insecurity” were found for Western Europe.	Poor diets and low breastfeeding rates are associated, highlighting how nutritional deficiencies can undermine maternal and child health practices [73].

SES: socioeconomic status.

## Data Availability

Data sharing is not applicable.

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
