# Peer review of "Maternal Socioeconomic Status and the Initiation and Duration of Breastfeeding in Western Europe Versus Southern Africa: A Systematic Review—A Contribution from the ConcePTION Project"

_nutrients, 2025, doi:10.3390/nu17060946_

Round 1
Reviewer 1 Report
Comments and Suggestions for Authors
The aim of this systematic review is to explore the socioeconomic status (SES) indicators that affect breastfeeding initiation and duration in two United Nations-defined regions: Western Europe and Southern Africa.
The paper is well-written, however, there are some suggestions to improve it:
Introduction:
I think it is worth mentioning how SES not only affects breastfeeding but also has long-term implications for child and maternal health, including nutritional status, immunity, and cognitive development. It is also related to the level of education, which relates to breastfeeding.
Methods:
1. While the databases used are mentioned, it would be helpful to include details on specific search terms and Boolean operators.
2. Clarify whether inter-reviewer agreement was measured (e.g., Cohen’s kappa) and how disagreements were resolved beyond involving a third reviewer.
Results:
The results mention that regional and environmental factors influence breastfeeding (e.g., urban vs. rural settings), but further elaboration on how these interact with SES would provide a more comprehensive picture.
Discussion:
The discussion suggests targeted interventions but could be more specific. For instance, what types of workplace policies have been successful in promoting breastfeeding? What community-based programs have shown positive effects in lower-income settings?
Author Response
Dear reviewer 1,
Many thanks for your time and effort invested reviewing our manuscript. We have included our responses below and the corresponding revisions in the resubmitted file.
The aim of this systematic review is to explore the socioeconomic status (SES) indicators that affect breastfeeding initiation and duration in two United Nations-defined regions: Western Europe and Southern Africa.
The paper is well-written, however, there are some suggestions to improve it:
Introduction:
I think it is worth mentioning how SES not only affects breastfeeding but also has long-term implications for child and maternal health, including nutritional status, immunity, and cognitive development. It is also related to the level of education, which relates to breastfeeding.
Methods:
While the databases used are mentioned, it would be helpful to include details on specific search terms and Boolean operators.
We agree that the specific search terms are helpful to be included in the paper, while somewhat distractive for the full paper. However, the search terms for each individual database were added as supplementary file S1, as mentioned in line 121, now line 144. We have further stressed this.
Clarify whether inter-reviewer agreement was measured (e.g., Cohen’s kappa) and how disagreements were resolved beyond involving a third reviewer.
The inter-reviewer agreement was not logged. If the 2 researchers did not agree, the third reviewer was consulted for their independent decision, resulting in a majority vote of 66.6% as mentioned in line 144-145, lines 164-165 in revised version. That’s in line with the PRISMA guidance and the practices on systematic reviews. We have provided additional information on this aspect in the revised version of the paper.
Results:
The results mention that regional and environmental factors influence breastfeeding (e.g., urban vs. rural settings), but further elaboration on how these interact with SES would provide a more comprehensive picture.
We have added some information in e.g. the discussion, and further refer to the detailed supplement that discusses all specific details of all the retained studies. In this way, we try to find a balanced between readability of the paper, and accessibility to all information collected.
Discussion:
The discussion suggests targeted interventions but could be more specific. For instance, what types of workplace policies have been successful in promoting breastfeeding? What community-based programs have shown positive effects in lower-income settings?
Some examples were provided in the paragraph from line 359-380 (revised lines 407 onwards) to clarify potential interventions, such as the Baby-Friendly Hospital Initiative.
However, the Baby-Friendly Hospital and Community Initiatives could be expanded and improved so more hospitals and communities are reached. Other possible interventions or policy changes, more specifically for the work environment, are workplace accommodations or (prolonged) paid maternity leave, as this is not always available in countries in Southern Africa.

Reviewer 2 Report
Comments and Suggestions for Authors
- The title about “A contribution from the ConcePTION project” was a little confusing. Did it matter to the topic? What did it mean?
- The abstract section did not make readers catch clear results or conclusions about the contrasts between “West Europe versus Southern Africa.” It was also difficult to obtain a simple understanding of the influencing direction of each SES factor.
- Finally, 47 papers/studies were selected for the “Materials and Methods” section. How many studies belong to West Europe and Southern Africa, respectively? Maybe the numbers could also be reported.
- Could the authors offer a Table in the final “Result” section to summarize the results based on two dimensions, one for “West Europe versus Southern Africa” and the other for SES factors? That would make your contribution clearer and would promote the manuscript being cited in the future.
Author Response
Review 2
Dear reviewer 2,
We thank you for your time reviewing our manuscript. Please find our responses below and the corresponding revisions in the resubmitted file.
The title about “A contribution from the ConcePTION project” was a little confusing. Did it matter to the topic? What did it mean?
An explanation regarding the ConcePTION project and its relevance was added to the introduction of this manuscript to add information on this link.
The abstract section did not make readers catch clear results or conclusions about the contrasts between “West Europe versus Southern Africa.” It was also difficult to obtain a simple understanding of the influencing direction of each SES factor.
Since the interrelationship of SES factors and breastfeeding practices is complex, there is not always a clear direction of the influence of each SES factor. Many SES factors show mixed effects on breastfeeding initiation and duration, such as employment and income status or community, regional, and religious factors in West Europe. We tried to highlight the main differences between West Europe and Southern Africa in this abstract.
Finally, 47 papers/studies were selected for the “Materials and Methods” section. How many studies belong to West Europe and Southern Africa, respectively? Maybe the numbers could also be reported.
Out of the 47 included articles, 29 studies belong to Westen Europe, and 18 studies belong to Southern Africa. The countries (and the region) reported in the included articles are mentioned in results, section 3.1. We do agree that its place was not ideal and have changed this in the revised manuscript.
Could the authors offer a Table in the final “Result” section to summarize the results based on two dimensions, one for “West Europe versus Southern Africa” and the other for SES factors? That would make your contribution clearer and would promote the manuscript being cited in the future.
A table has been added to the discussion section to summarize the results of this review. Thank you for this suggestion, and we assess this to be of add on value to the paper.

Round 2
Reviewer 2 Report
Comments and Suggestions for Authors
line 322, Table 1 or Table 4? a typo error